# The Qualitative Characteristics of Accounting Information, Earnings Quality, and Islamic Banking Performance: Evidence from the Gulf Banking Sector

Ibrahim Elsiddig Ahmed

Department of Accounting, College of Business Administration, Ajman University, Ajman 346, UAE;
i.alsiddiqe@ajman.ac.ae

**Abstract:** The study aims to operationalize financial reporting quality in terms of the qualitative characteristics (QCs) as stated by the Accounting and Auditing Organization of Islamic Financial Institutions (AAOIFI) standards, as well as to investigate their association with earnings quality (EQ) and banking performance. The study uses secondary data extracted from DataStream to operationalize and measure the financial reporting quality in the annual reports of 25 out of the 27 Islamic banks in the Gulf Council Countries (GCC) for a 5-year period (2014–2018), meaning 125 annual reports were used. The study applies a manual content analysis to the annual reports to score all the items of QCs and operationalizes 25 measurement items that represent the six QCs. All items use 5-point Likert-type scales to compute the sub-score and the overall index through the Neural Network System. The findings of the model paths show a significant positive relationship between EQ and most of the QCs. The first hypothesis is partially accepted as there is a positive relationship between EQ and relevancy, reliability, prudence and general quality; however, there is no significant relationship between EQ and understandability and there is a significant negative relationship between EQ and comparability. Moreover, the study finds a significant positive relationship between EQ and ROA on one hand and EQ and ROE on the other hand (*p*-value = 0.00), meaning the second hypothesis is supported.

**Keywords:** qualitative characteristics; earnings quality; AAOIFI conceptual framework; Islamic banking performance; M40; N25

## 1. Introduction

The rapid growth in Islamic banks followed by a significant demand for their services lead to a huge need not for the amount of presented information but a need for high quality. The research conducted in the area of Islamic banking finance and reporting was based on comparative analysis between Islamic and conventional banks. Duha (2016) compared earnings management in Islamic banks with conventional banks. This study investigates the reporting quality and performance only in Islamic banks and not applying comparative analysis with conventional banks. Many previous studies have provided evidence on the positive role of financial reporting quality as well as its influence on the performance and quality of decisions (Jaballah et al. 2014; Lin et al. 2016). While most studies concentrate only on relevancy and reliability, this study considers all the qualitative dimensions of information and their association with earnings quality (EQ). Numerous previous studies have explored determining factors for financial reporting quality in developed countries, such as the USA, the UK and Canada (Monday and Nancy 2016). Meanwhile, most of the studies on financial reporting quality have focused on non-financial companies (Abdul Majid and Ismail 2008; Thalassinos et al. 2015; Vovchenko et al. 2017). Few studies in this field have been carried out in the context of financial

institutions. This study tries to bridge this gap as it concentrates on the reporting quality of Islamic banks, which were not studied before.

The preparation and presentation of quality financial reports have positive impacts on useful financial and management decisions. The usefulness of financial information relies on its qualitative characteristics (QCs), which are stated according to Accounting Standards (FASB, IAS, IFRS, AAOIFI). The main indicators of financial information quality from the perspective of the developers of accounting standards are relevance and reliability, as these make information useful for decision-makers (Nwaobia et al. 2016). Sutan and Nayla (2014) studied the compliance of financial reports in Bahrain Islamic banks to the AAOIFI requirements. Sarea (2012) also studied the level of compliance with AAOIFI Accounting Standards in Bahrain about the perceptions concerning the level of compliance with AAOIFI Accounting Standards. His study indicated that Islamic banks of Bahrain have fully adopted the AAOIFI Accounting Standards. The evidence of high compliance to the standards by previous studies was one of the motivations for the author to investigate the quality of information prepared by Islamic banks not only in Bahrain but in all the six Gulf Council Countries (GCC). It should be noted that the special nature of Islamic banking products is likely to require a different quality of financial reports, as well as a special treatment of their profits. Most of the studies on Islamic banks concentrate on legal issues, products, profitability and efficiency (El Moussawi and Obeid 2010; Akhtar et al. 2011; Aljifri and Khandelwal 2013).

The reporting system of Islamic banks is different from that of conventional banks because of the differences in the objectives and the users' needs, as users desire high-quality information to ensure that the banks are following Sharia rules and guidelines (Ahmed 2012). Despite the importance of reporting quality, the areas of earnings and reporting quality in the context of Islamic banks and Islamic banking products have been neglected in the literature on Islamic finance. There are no previous studies on the quality of financial reports in the Islamic banking sector. Hence, this study contributes significantly to the literature of both Islamic finance and Islamic accounting. Also, the quality of financial reports can be assessed based on their usefulness to the users or on their fairness and completeness to the investors. This study is different in that it applies a comprehensive approach that combines both views to assess the quality of financial reports through the QCs of the AAOIFI framework. It is thus the first study to classify the QCs into six general categories (general, relevance, understandability, reliability, comparability and prudence). This represents a contribution to the literature and to practice through the development of a comprehensive measurement tool to assess the quality of financial reporting based on the AAOIFI QCs and the contents of Islamic banks' reports. Furthermore, most of the mentioned studies use information quality and EQ interchangeably, whereas this study assesses the quality of financial reports from both dimensions, namely the QCs of accounting information and the EQ. Finally, previous studies have highlighted the financial reporting determinants in developed economies (Takhtaei et al. 2014; Soheilyfar et al. 2014; Monday and Nancy 2016; Al-Asiry 2017; Pivac et al. 2017). However, it may not be possible to generalize their results to developing countries as these have distinct cultural, environmental, economic and political settings (Alfraih and Almutawa 2014; Kuznetsova et al. 2017).

This paper is organized as follows. Section 1 is an introduction that explains the justifications and objectives based on previous studies. Section 2 is about the literature on reporting quality, a brief overview of Islamic Banking in the GCC, earnings quality, and banking performance. Section 3 presents the sources of data, identifies the variables and operationalizes the QCs. Section 4 conducts analysis and presents the results and findings. The conclusion is stated in Section 5.

*Objectives of the Study*

The main objective of this study is to assess the quality of the financial reporting system in the Islamic banking sector. This study is expected to contribute to the measurement of the quality of financial reporting based on both QCs and EQ. Therefore, the study aims to operationalize the financial reporting quality in terms of the QCs as stated by the Accounting and Auditing Organization of Islamic Financial Institutions (AAOIFI) standards. The overall AAOIFI QCs are consistent with those of IFRS

or IASB, but they are also different from other standards in terms of the details and their unique definitions in addition to special contents. Moreover, the study aims to assess the relationship between the quality of financial reports and the quality of earnings, as currently measured on an accrual basis by Islamic banks. In addition, this study attempts to provide a further investigation into the relationship between reporting quality and the performance of GCC Islamic Banks.

## 2. Literature Review

The measurement of disclosure quality is useful for organizations because it offers the prospect to decrease the cost of debt and enhance the stock price (Soheilyfar et al. 2014; Savina 2016). Currently, the reporting system includes a wide range of contents that need to be disclosed to provide investors with vital information that they need to evaluate their investments, and this no longer limited to financial statements (Asegdew 2016; Liapis and Thalassinos 2013; Allegret et al. 2016).

The existence of many users of accounting information requires it to be viewed from different perspectives, which has created difficulty in measuring the QCs of financial information. The determinants influencing financial reporting quality are related to certain attributes of the firm, such as type of industry, leverage, firm size, firm age, profitability, liquidity, type and size of the audit firm and the status of listing (Soheilyfar et al. 2014; Al-Asiry 2017; Abdul Majid and Ismail 2008; Haji and Ghazali 2013; Chakroun and Hussainey 2014; Suryanto 2017). Other determinants that influence financial reporting quality include the features of corporate governance, such as board size and composition, the duality of CEOs, board meetings and ownership structure (Fathi 2013; Chakroun and Hussainey 2014; Liapis and Thalassinos 2013).

The Islamic banking industry started in the early 1970s and from that time until the late 1990s, their main concerns were about legalizing finance transactions. The market share of Islamic financing assets had increased in 2019 to 47% of total financial assets from 39% in 2016 (AAOIFI 2017). The increased market share was led by the GCC in which the number of Islamic banks jumped to 30 banks by the end of 2019. This growth of Islamic banks is supported by the customers' high demand for Islamic services as well as governmental support. Islamic banks offer Profit-Sharing investment accounts as Sharia-compliant deposits. Furthermore, Islamic Banks combine commercial banking activities and investment banking operations to enhance profitability but in compliance with Sharia rules and principles. As the demand for such services continues to grow, a question arises about the quality of presented information and its implication on users. Adequate disclosure about management and performance of investment accounts and other stakeholders is important for making proper investment decisions. Given the important role played by Islamic banks in the GCC market economy, since almost all financial transactions are being made through banks, and given the competitive market in which they are operating, it is important to study the quality of their reports and their profitability.

Islamic banks' activities are governed by secular law, the environment and control. AAOIFI was established for the development of standards for Islamic institutions throughout the world. The reporting system of Islamic banks is still at the stage of developing an understanding and presenting the unique nature of Islamic banking and determining whether the accounting treatment of Islamic banking products follows Sharia standards. It is high time to measure and prepare quality information to help the users of financial statements in taking quality decisions. In the GCC, the need for information from different users is different from that in developed countries, as their behavior is different. Dospinescu et al. (2019) stated that bank customers are different in behavior from one country to another in terms of profile.

### 2.1. Earnings Quality (EQ)

One popular definition of EQ is the absence of earnings management. EQ can be explained by the relevancy and reliability of financial statements as perceived by the users. So far, there is no common definition of EQ, but it is assessed based on certain indicators, such as accrual basis and quality, predictability, smoothness, persistency, timeliness, conservatism and earnings uncertainties.

Jing (2007) found a positive association between accrual quality and stock price synchronicity and an insignificant relationship between conservatism and stock price synchronicity.

EQ can be generally classified into two main types: The first is accountability, also known as decision usefulness and stewardship, which is based on accrual quality and conservatism, where EQ is measured based on the future persistence of current earnings. The second is value relevance, which reflects the explanatory power of earnings and the book value of equity for stock returns.

Many studies have documented that poor-quality information, particularly earnings, is the reason for increasing information asymmetry (Bhattacharya et al. 2013; Lara et al. 2011). Consequently, Steven et al. (2009) conducted a study to analyze the association between returns and EQ for 13 countries.

Aljifri and Khandelwal (2013) explain that "Contracts in conventional financial systems are purely drawn and based on material information, facts and conditions, whereas the contracts in the Islamic financial system are made of material and ethical components". This approach suggests the development of an index that is composed of the elements of the income statements. There are different measures of EQ. One simple approach is the ratio of operating cash flow to income; this measure of EQ assumes that a high ratio means high quality. The variability of earnings between cash and accrual, which is computed by Leuz et al. (2003), by comparing the standard deviation percentage of income from operations to the standard deviation of operating cash flow. Barton and Simko (2002) used a technique base on earnings surprise measured by comparing the beginning balance of net operating assets to sales revenues. The variability of earnings between cash and accrual, as applied by Leuz et al. (2003), measures the variability of earnings by calculating the ratio of the standard deviation of operating earnings to the standard deviation of cash from operations. Barton and Simko (2002) developed an approach focusing on earnings surprise as reflected in the beginning balance of net operating assets relative to sales. Chan et al. (2006) studied the relationship between EQ and stock returns in the USA and found that poor EQ is associated with poor future returns. Anderson et al. (2014) examined the relationship between corporate social responsibility, earnings management and firm performance, indicating that there is a negative relationship between corporate social responsibility and earnings management, while there is no relationship between corporate social responsibility and firm performance. Farooq and AbdelBari (2015) studied the difference in EQ between Shariah-compliant firms and other firms. Their main finding was that Shariah-compliant firms engage in lower earnings management as compared to other firms.

Therefore, this study tries to investigate the relationship between the contents of the QCs of the AAOIFI framework and EQ. Based on the above discussion of the previous literature, the study develops the following hypothesis:

**Hypothesis 1 ($H_1$).** *There is a significant positive relationship between the QCs of financial reports and EQ.*

Because the QCs have different components, the study segregates this hypothesis into the following sub-hypotheses:

**Hypothesis 1a ($H_{1a}$).** *There is a significant positive relationship between EQ and reliability.*

**Hypothesis 1b ($H_{1b}$).** *There is a significant positive relationship between EQ and relevancy.*

**Hypothesis 1c ($H_{1c}$).** *There is a significant positive relationship between EQ and general quality.*

**Hypothesis 1d ($H_{1d}$).** *There is a significant positive relationship between EQ and understandability.*

**Hypothesis 1e ($H_{1e}$).** *There is a significant positive relationship between EQ and comparability.*

**Hypothesis 1f ($H_{1f}$).** *There is a significant positive relationship between EQ and prudence.*

### 2.2. The Performance

As financial reporting quality is expected to lead to better and more useful decisions, it may result in better financial performance. The performance of Islamic banks is necessary to assess the soundness and the safety of the banking system, maintain the public confidence in the banking system and identify the banks in financial distress. Olson and Zoubi (2011) applied some performance indicators based on profitability ratios in the Middle East and North Africa (MENA) banking sector and found that banks in developing countries are similarly efficient. Although MENA banks are relatively profit-efficient, investors face many difficulties when trying to make optimal decisions. Some studies indicated that investors are not fully satisfied with the profits reported by companies operating in the MENA region and they attribute that to the low quality of reported information (Bahrini 2017).

Bashir (2001) measured Islamic banking profitability through Non-Interest Margin (NIM), Earning before Tax (EBT), Return on Assets (ROA) and Return on Equity (ROE). His results show that Islamic Banks' profitability is positively related to equity and loans. All the previous measures did not explain the impact of reporting quality on performance. Uyen (2011) applied the CAMEL approach to assessing banking performance, which is composed of five variables (Capital, Assets, Management, Earnings and Liquidity) and helped avoid banks' failure and inefficiencies by anticipating any potential risks and taking preventive actions. In the literature, some researchers examined the impact of CAMEL's five components on banks' profitability using ROA and ROE as proxies of performance (Debnath 2017).

Many researchers have investigated the real benefits of the quality of financial reporting. For example, Lin et al. (2016) indicated that inadequate financial reporting might negatively influence business performance and decision-making. The basic measures of performance are the profitability indicators. Meanwhile, there is varying evidence supporting the association between a firm's profitability and its financial reporting quality. Raffournier (2006); Dedman et al. (2008); Fathi (2013); Uyar et al. (2013); Takhtaei et al. (2014) and Al-Asiry (2017) found a significant positive relationship between profitability and financial reporting quality. Dospinescu and Dospinescu (2019) noticed a positive relationship between the net profit margin and stockholder's equity but have a negative relationship with working capital. The firm's performance measures are extracted from the financial reports that need to be of high quality to be useful for investors. The measures of business performance are either absolute values in the form of net income or the form of different ratios when comparing profits to equity, assets or revenues. Bill et al. (2016) found a significant relationship between profit margins and peer quality. Other studies found a positive relationship between firms' growth opportunities and the quality of their earnings disclosure (Fathi 2013). On the other hand, some researchers who did not find a relationship—or only found an insignificant relationship—between EQ and business performance, such as Sadeghi and Zareie (2015) investigating the relationship between earnings management and the financial ratios of private companies listed on the Tehran Stock Exchange, Debnath (2017) analyzing the impact of business growth and efficiency on earnings management in India. In addition, Aggarwal et al. (2019) found a relationship between performance and quality of corporate governance in the form of board diversity, while Jang and Kim (2017) studied the influence of financial indicators on earnings management in the Korean ready-mix concrete industry. Meanwhile, Monday and Nancy (2016) and Ebrahimabadi and Asadi (2016) proved a negative relationship between profitability and disclosure quality. Therefore, this study formulates the following hypothesis:

**Hypothesis 2 (H₂).** *There is a significant relationship between the profitability and EQ of GCC Islamic banks.*

### 3. The Methodology

### 3.1. Data Collection and Analysis

This study operationalizes and measures the financial reporting quality in the annual reports of 25 out of the 27 Islamic banks in the Gulf Council Countries (GCC) (93% of the total population) for 5 years (2014–2018), thereby using 125 annual reports. Two Islamic banks are excluded because they

do not have comprehensive reports on DataStream. Islamic banks in the GCC own more than 60% of the Islamic banks' assets and have been proven to have the highest growth rates and the highest efficiency as compared to their counterparts in other parts of the world, as reported by Bahrini (2017).

Financial reporting quality is measured on different bases in the literature (Pivac et al. 2017). Previous studies applied many measures, such as accrual models (Van Tendeloo and Vanstraelen 2005) or value relevance models (Barth et al. 2001; Nichols and Wahlen 2004) and research has also focused on certain contents in the annual report (Mbobo and Ekpo 2016), as well as methods for operationalizing QCs (Barth et al. 2008; Van Beest et al. 2009). Although many measurement techniques have been employed in the literature to assess the quality of financial reporting, this study is based on the operationalizing method of the QCs of accounting and financial information.

This study uses a quality index for annual reports, adopted from Braam and Beest (2013) with some modifications. These indices are selected because they are capable of comprehensively assessing the quality of financial and non-financial information in the annual reports while considering all aspects of decision usefulness, as interpreted in the Conceptual Framework for Financial Reporting of AAOIFI. The major task of the development of the disclosure index is built around the selection of the elements to include in the index (Marston and Shrives 1991).

The independent variable of the study is the overall quality score of accounting information, which is composed of the six elements of the QCs as prescribed by AAOIFI. The study measures the quality score on a three-layer base that starts from the sub-components of each element, then moves to the second layer of the main six elements and finally to the last layer of the overall quality score.

To measure each level of the score's layers, the study applies a manual content analysis to the annual reports. To control for biased interpretations, two research assistants provided individual second assessment scores for the QCs to determine the items' measurement scores. Krippendorff's alpha coefficient is used to guarantee the reliability of the scores.

The quality is operationalized by 25 measurement items for the following QCs: high general qualities (3 items, Q), relevance (4 items Rs), understandability (4 items, U), reliability (6 items, B), comparability (4 items, C) and prudence (3 items, P). All items use 5-point Likert-type scales. More details on the qualitative items are presented in Table 1.

Second, a sub-score is calculated for each QC of each annual report (sub-score= total score on the related items for each QC divided by the total number of items for each QC). Third, an aggregated score is calculated for the quality of each annual report (aggregated score = sum score of all related items of the QCs divided by 25).

The dependent variables: The study aims to investigate the relationship between the quality of financial reports, with EQ as the dependent variable. The study also considers a comprehensive concept of EQ that includes an absence of earnings management in addition to reliability and relevance and also considers the values and principles of Islamic banks to extend the meaning of EQ to cover issues involved in meeting certain standards (i.e., the level of compliance with Sharia law, ethics and societal needs).

The second dependent variable is profitability, which has common measures in the literature. This study selected two profitability measures: Return on Equity (ROE) = Net Income/Shareholders' Equity and Return on Assets (ROA) = Net Income/Total Assets. The selected measures were applied in previous studies such as Yao et al. (2018).

The study analyzes the collected data using the Neural Networks (NNW) because this system outperforms other approaches concerning one or more dependent variables where the form of the relationship between them is unknown (Du and Swamy 2006). Neural Networks provide a mechanism for modeling relationships between several inputs and one or more outputs; they act as a black box between the input and output. The study applies the most commonly used neural network architecture, which is the multilayer perceptron (MLP) neural network trained with a backpropagation rule.

**Table 1.** Operationalization of the Qualitative Characteristics (QCs).

| Qualitative Category | Definition | Criteria/Measures of the Category |
|---|---|---|
| **General Quality (GQ)** | The conceptual framework develops objectives and concepts that lead to high-quality accounting standards and financial accounting processes, which in turn also lead to high-quality financial reporting information that is useful for making decisions. | Q1: True and Fair Value, Q2: Decision Usefulness Q3: Transparency |
| **Relevance (R)** | Relevance refers to the existence of a close relationship between the financial accounting information and the objectives for which it was prepared. The relevance of that information to one or more decisions made by users is an indicator of usefulness. | R1: Predictive Value R2: Feedback Value R3: Timeliness R4: Frequency and Length |
| **Understandability (U)** | Understandability is the ability to know the comprehensive meaning of information. It is enhanced through proper classification as well as concise and clear presentation. It depends on the contents of financial statements and the presentation style as well as the background and abilities of users. | U1: Classification that is meaningful to users U2: Well-organized juxtaposition of data U3: Clear and sufficient notes U4: Presentations of net figures that users need |
| **Reliability (RL)** | Reliability is the characteristic that allows users to confidently depend on information. Reliability means that the method selected to measure and/or disclose its effects produces information that reflects the substance of the event. As per Sharia principles, estimates and judgments in applying accounting methods are not preferred. | B1: Representational Faithfulness B2: Neutrality B3: Substance and Form B4: Completeness B5: Verifiability B6: Consistency |
| **Comparability (C)** | Comparability allows users to identify similarities and differences among institutions and over a time horizon. | C1: Comparison of the current with previous financial periods C2: Comparison with other banks or industries C3: Analysis and explanation of the implications of changes C4: Presentation of ratios, index scores, or benchmarks |
| **Prudence (P)** | Prudence is needed under uncertainty to reflect the inclusion of a degree of caution in the exercise of the judgments needed in making the estimates, such that assets or income are not overstated and liabilities or expenses are not understated. | P1: Present reasonable information about uncertainties P2: Indicate bases of judgment P3: No hidden reserves or more provisions |

Source: Operationalized by the author based on the Accounting and Auditing Organization of Islamic Financial Institutions (AAOIF framework.

### 3.2. Qualitative Characteristics of Accounting Information

The QCs of accounting information refer to the qualities of useful accounting information and the basic principles used to assess its quality. The definition of these qualities should assist the bodies responsible for setting the accounting standards, as well as accountants who prepare financial

statements to evaluate the financial information produced by alternative accounting methods and in differentiating between necessary and unnecessary disclosures. Mbobo and Ekpo (2016) and Yurisandi and Puspitasari (2015) also stress the importance of QCs for assessing the information quality provided by financial reports. Table 1 explains the meanings and measures of QCs.

To test the hypotheses, the study analyzes the relationships between the six qualitative characteristics of accounting information and EQ on the one hand and between EQ and Islamic banking performance (ROA and ROE) on the other hand. The model that explains this relationship is presented in Figure 1 below.

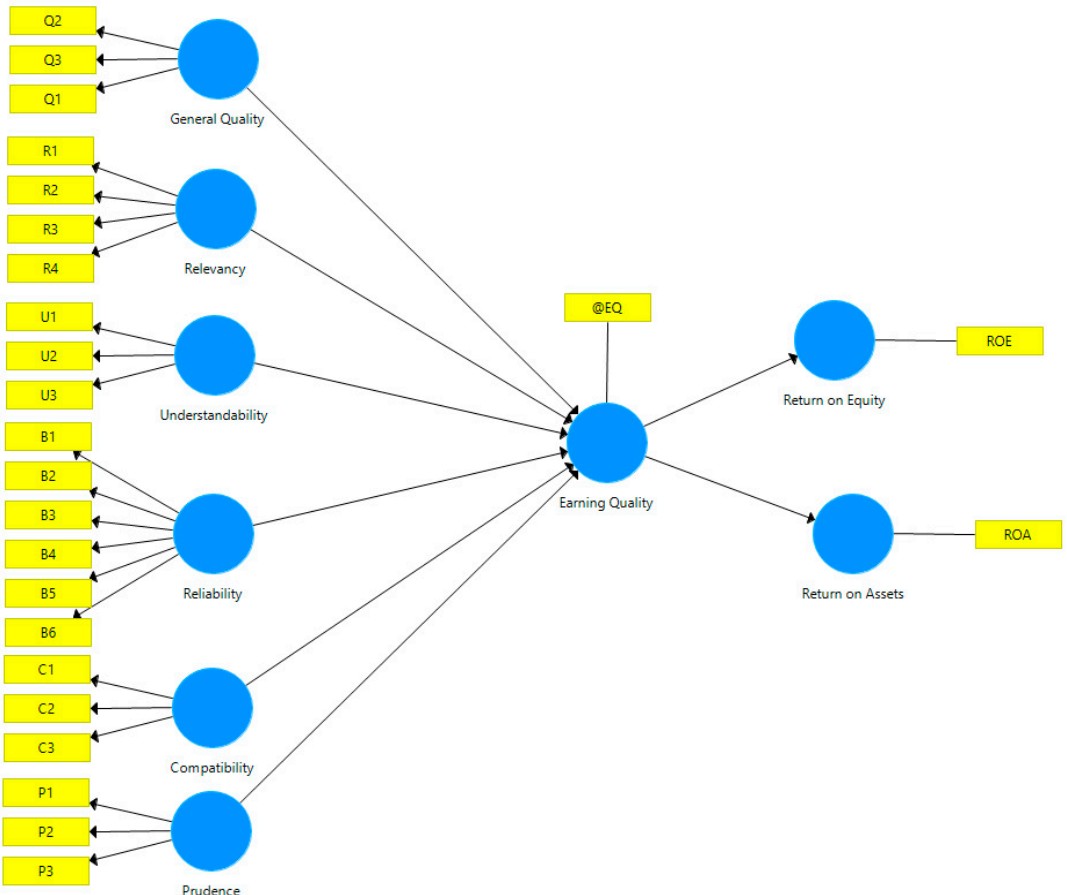

**Figure 1.** Research Model. Source: Developed by the author using the Neural Network System.

## 4. Results and Analysis

To measure the goodness of fit of the data, the study computed the coefficient of determination ($R^2$) and the adjusted $R^2$. Deltuvaitè (2010) stated that an R-square of 10% is generally accepted for studies in the field of arts, humanities and social sciences. Therefore, an $R^2$ of 0.10 or more for EQ, ROA and ROE is considered acceptable and indicates the goodness of the research model, as presented in Table 2.

**Table 2.** $R^2$.

|  | $R^2$ | Adjusted $R^2$ |
|---|---|---|
| Earnings Quality | 0.4982 | 0.4727 |
| Return on Assets | 0.1470 | 0.1400 |
| Return on Equity | 0.1849 | 0.1783 |

Source: Computed by the author (NNW).

The assessment of discriminant validity is the extent to which the constructs are empirically different from one another. It was developed by Henseler et al. (2015) and is useful in research that involves latent variables along with the use of many indicators to represent construct. This study measures discriminant validity through both the Fornell-Larcker method and the Heterotrait-Monotrait (HTMT) ratio of correlations method. This is to ensure that the latent constructs used for measuring the causal relationships are different and are not measuring the same thing, which would lead to the issue of multi-collinearity. Nine latent constructs (comparability, general quality, prudence, relevancy, reliability, understandability, EQ, return on assets and return on equity) have been measured by this study and each one includes sub-layers that measure it. The Neural Network is used to compare and analyze both methods of discriminant validity. Table 3 below highlights the reliability and validity results of the study. The output from the analysis of the composite reliability and the average variance extracted (AVE) are summarized in Table 3.

**Table 3.** Construct Reliability and Validity.

|  | **Cronbach's Alpha** | **Rho A** | **Composite Reliability** | **Average Variance Extracted (AVE)** |
|---|---|---|---|---|
| Comparability | 0.6844 | 0.7337 | 0.8121 | 0.5944 |
| General Quality | 0.7335 | 0.7560 | 0.8476 | 0.6500 |
| Prudence | 0.7402 | 0.7560 | 0.8496 | 0.6535 |
| Relevancy | 0.8257 | 0.8500 | 0.8839 | 0.6571 |
| Reliability | 0.8450 | 0.8567 | 0.8844 | 0.5620 |
| Understandability | 0.7232 | 0.7343 | 0.8422 | 0.6402 |

Source: Computed by the author (NNW).

Cronbach's alpha is the most common measurement for internal consistency and composite reliability. It measures the reliability based on the interrelationship of the observed items' variables. The internal consistency among the variables is achieved and acceptable once the values of Cronbach's alpha reach 0.70. The first column of Cronbach's alpha in Table 3 proves that all QCs are acceptable because their scores are above 0.7 except comparability (0.68), which lies marginally below the acceptable target.

The composite reliability of each item is measured by the cross-loadings presented in column 3 of Table 3. The values of composite reliability are high on their respective constructs, which means that each factor loading is greater than the cut-off value of 0.70. It is an indicator that the reliability of each variable is good, reinforcing to the allocation for each item on the specified latent construct. To prove differences among the different the QCs, the study compares the square root of each AVE. The findings show that their values are within 0.56 and 0.65, which indicates very small differences. It is an acceptable validity score, as column 4 of Table 3 above shows that all the AVE values are above 0.50.

The Fornell-Larcker method computes the correlation coefficients of the latent constructs, as shown in Table 4. When comparing the results from Tables 3 and 4, it becomes clear that that the square root of each construct's AVE is greater than the correlations with other latent constructs. Therefore, discriminant validity can be accepted for this measurement model and this supports the discriminant validity between the constructs.

Analyzing the results of the Heterotrait-Monotrait Ratio (HTMT) in Table 5, only one value indicates a discriminant validity problem according to the HTMT (0.85) criterion. This implies that the HTMT criterion is detecting a collinearity problem among the latent constructs (multi-collinearity) between general quality and relevancy (0.88). This means that general quality and relevancy are measuring the same thing or that both constructs overlap.

**Table 4.** Fornell-Larcker Criterion.

| | Comparability | Earnings Quality | General Quality | Prudence | Relevancy | Reliability | Understandability |
|---|---|---|---|---|---|---|---|
| Compatibility | 0.7710 | | | | | | |
| General Quality | 0.3736 | 0.5577 | 0.8063 | | | | |
| Prudence | 0.6078 | 0.50067 | 0.4304 | 0.8084 | | | |
| Relevancy | 0.3220 | 0.5481 | 0.6812 | 0.4699 | 0.8106 | | |
| Reliability | 0.4013 | 0.4965 | 0.4880 | 0.4835 | 0.3795 | 0.7497 | |
| Understandability | 0.0062 | 0.2731 | 0.3517 | 0.2367 | 0.287 | 0.1913 | 0.8001 |

Source: Computed by the author (NNW).

**Table 5.** Heterotrait-Monotrait Ratio (HTMT).

| | Comparability | Earnings Quality | General Quality | Prudence | Relevancy | Reliability | Understandability |
|---|---|---|---|---|---|---|---|
| Compatibility | | | | | | | |
| General Quality | 0.5432 | 0.6310 | | | | | |
| Prudence | 0.8399 | 0.5625 | 0.5719 | | | | |
| Relevancy | 0.4226 | 0.5916 | 0.8829 | 0.5952 | | | |
| Reliability | 0.4886 | 0.5193 | 0.5982 | 0.6106 | 0.4501 | | |
| Understandability | 0.1084 | 0.3138 | 0.4797 | 0.3181 | 0.3914 | 0.2457 | |

Source: Computed by the author (NNW).

*Assessing the Measurement Model Fit*

Table 6 presents the analysis of the model fit by assessing several indices to ascertain the degree of the model fit. As indicated in Table 6, five indices (SRMR, d-ULS, d-G, chi-square and NFI) are applied to assess the fitness of the model. The Standardized Root Mean Square Residual (SRMR) is based on transforming both the sample covariance matrix and the predicted covariance matrix into correlation matrices. In both, the value of SRMR is 0.10 or less, which proves that the model is acceptable and has a good fit. As defined by Dijkstra and Henseler et al. (2015), d_ULS (the squared Euclidean distance) and d-G (the geodesic distance) represent two different ways to compute this discrepancy. The model fits well as the difference between the saturated and estimated models is very small. The Normed Fit Index (NFI) also proved to some extent that the model is fit because it has a value above 0.50. The chi-square is large because of the model size, which also proves the high fitness of the model.

**Table 6.** Model Fit.

| Fit Summary | Saturated Model | Estimated Model |
|---|---|---|
| SRMR | 0.0841 | 0.1034 |
| d_ULS | 2.2982 | 3.4713 |
| d_G | 0.9547 | 1.1835 |
| Chi-Square | 638.7540 | 748.1588 |
| NFI | 0.6249 | 0.5606 |

The examination of the individual model paths shows a significant relationship between EQ and most of the QCs of financial information. Only the quality of understandability is not significantly related to EQ because the *p*-value is 0.95, which is above 0.05, and the t-statistic is 0.06, which is very low. Therefore, the first hypothesis is accepted because there is a positive relationship between EQ on the one hand and relevancy, reliability, prudence and general quality on the other hand. This supports the acceptance of hypotheses $H_{1a}$, $H_{1b}$, $H_{1c}$ and $H_{1f}$, but hypotheses $H_{1d}$ and $H_{1e}$ must be rejected as there is no significant relationship between EQ and understandability and there is a significant negative relationship between EQ and comparability. The low scores of the sub-components of the understandability (U1 = 10.5, U2 = 7.4, and U3 = 9.3) is the main reason for a non-supporting relationship with EQ.

The results of Table 7 provide support for a significant positive relationship between EQ and banking performance because there is a positive significant relationship between EQ and ROA on the one hand and EQ and ROE on the other hand (*p*-value = 0.00). This result fully supports the second hypothesis (H2: There is a significant relationship between the profitability and earnings quality of GCC Islamic banks). The findings of this study are consistent with Ahmed and Duellman (2011), who found the quality of financial reporting has a positive effect on the overall higher performance of the company.

**Table 7.** Hypotheses Testing.

|  | Sample Mean (M) | Standard Deviation (STDEV) | T Statistics (\|O/STDEV\|) | *p*-Value | Decision |
|---|---|---|---|---|---|
| Comparability -> Earnings Quality | −0.2292 | 0.0965 | 2.9774 | 0.0029 | Supported |
| Earnings Quality -> Return on Assets | 0.3858 | 0.0527 | 7.2714 | 0.0039 | Supported |
| Earnings Quality -> Return on Equity | 0.4305 | 0.0749 | 5.7376 | 0.0017 | Supported |
| General Quality -> Earnings Quality | 0.2359 | 0.1124 | 2.2551 | 0.0242 | Supported |
| Prudence -> Earnings Quality | 0.3176 | 0.0909 | 3.8721 | 0.0001 | Supported |
| Relevancy -> Earnings Quality | 0.2214 | 0.1099 | 1.9439 | 0.0419 | Supported |
| Reliability -> Earnings Quality | 0.2369 | 0.0820 | 2.8975 | 0.0038 | Supported |
| Understandability -> Earnings Quality | 0.0158 | 0.0703 | 0.0625 | 0.9502 | Not Supported |

Figure 2 shows the model components and their relationships. This model explains how EQ is explained by the QCs of accounting information. The statistical cut-off point is the score of 1.96. Out of the six quality variables, understandability is only 0.062, which is far below 1.96 in supporting EQ, while the majority of the QCs variables are above 1.96 and support the value of EQ. The relevancy score is 1.94, which falls close to 1.96 and this explains that it is a marginally supportive variable.

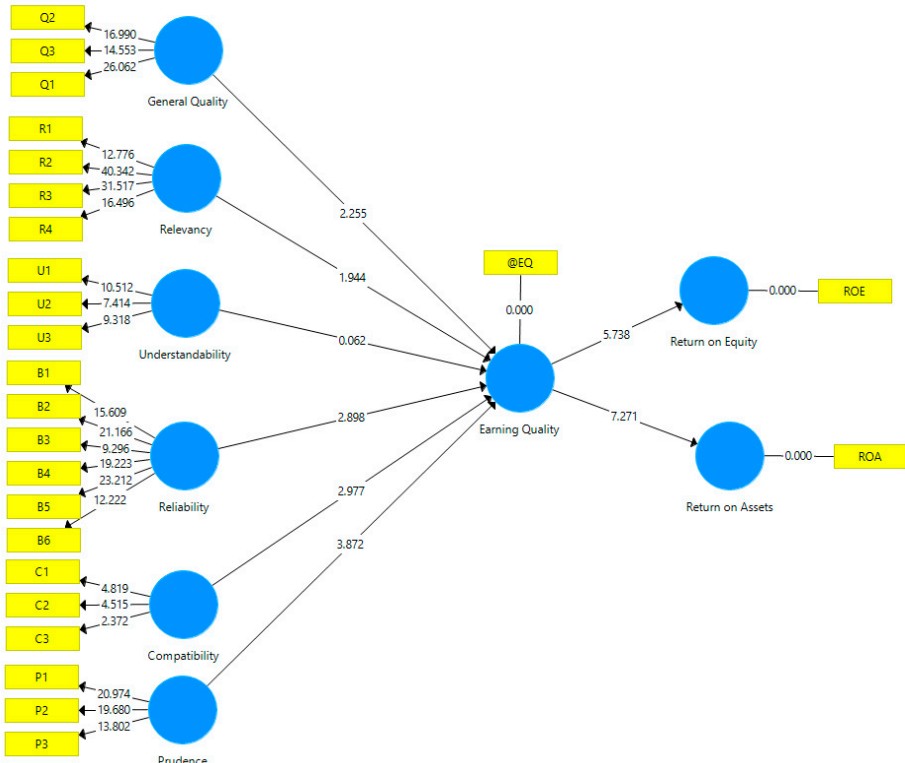

**Figure 2.** Research Model with Hypotheses Testing. Source: Developed by the author based on software results.

Figure 3 ranks the degree of support of the QCs for EQ and EQ for the performance elements (ROE and ROA). The prudence characteristic ranks first concerning its support for EQ (0.352), followed by general quality (0.254), reliability (0.238) and relevancy (0.214). Meanwhile, there is a significant negative relationship between comparability and EQ (−0.287) and understandability has a negligent impact of almost zero. EQ explains 18.5% of ROE and 14.7% of ROA and positively supports both performance measures (ROE 0.43 and ROA 0.383). The findings of this analysis indicate that all hypotheses should be accepted except for the two hypotheses of understandability and comparability. Therefore, the study rejects hypothesis $H_{1d}$ as there is no significant relationship between EQ and understandability and $H_{1e}$ because there is a significant negative relationship between EQ and comparability.

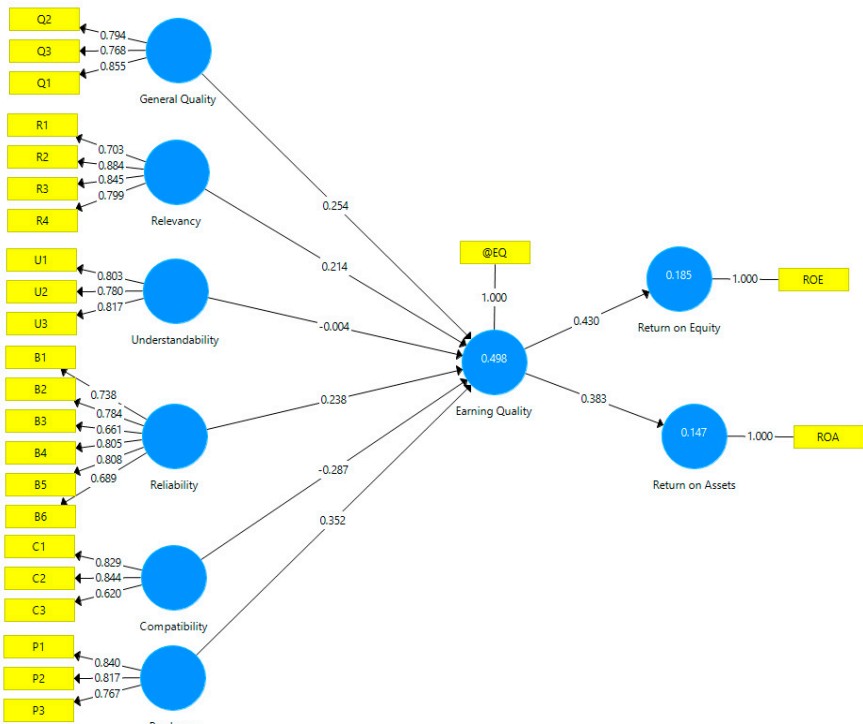

**Figure 3.** Research Model with Outer Loads and Beta. Source: Developed by the author based on software results.

## 5. Conclusions

The rapid growth in Islamic Banking has generated an increased interest in understanding the quality of their reports and their impact on earnings quality and performance. All previous studies concentrate on comparison with conventional banks at different levels. The stated objective of the study is the measurement of qualitative characteristics of accounting information and explain its impact on earnings quality and profitability of Islamic banks in the GCC. To achieve stated objectives and test the hypotheses, the study collects data from the annual reports of 25 out of 27 Islamic banks in the GCC over five years (20 quarters), we applied the Neural Network System to analyze the data and test two main hypotheses, whereby the first one was composed of six sub-hypotheses. Out of the total of seven hypotheses, only two were rejected. It is the first study that operationalizes and measures the QCs of accounting information as stated by the AAOIFI of Islamic Banks in the GCC. The six QCs have been operationalized into twenty-two sub-components based on AAOIFI conceptual framework that ends up with a quality index. Further, it investigates the relationship between the QC index and EQ, as well as banking profitability. The study finds a positive relationship between EQ on the one hand and relevancy, reliability, prudence and general quality on the other hand. Furthermore, no significant relationship between EQ and understandability was found, while there is a negative

significant relationship between EQ and comparability. To achieve the second objective, the study finds that there is a positive significant relationship between EQ and ROA on one hand and EQ and ROE on the other hand.

The study contributes to the literature and practice of Islamic banks. It also adds to the body of knowledge and helps managers and investors to recognize the QCs and their impact on decisions and profitability. The study recommends further studies to investigate whether findings apply to Islamic banks in countries outside the GCC. One of the limitations is only covering Islamic banks in a limited geographical area and evaluating performance on only two indicators. Therefore, the study recommends including all the CAMEL variables as performance measures.

**Funding:** This research received no external funding.

**Acknowledgments:** The author thanks Anshuman Sharma for his kind support in data validation and analysis.

**Conflicts of Interest:** The author declares that there is no conflict of interest.

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
