# Peer review of "The Qualitative Characteristics of Accounting Information, Earnings Quality, and Islamic Banking Performance: Evidence from the Gulf Banking Sector"

_ijfs, doi:10.3390/ijfs8020030_

Round 1

Reviewer 1 Report

Abstract, title and references

The purpose is clear. It is clear what the study discovered and how they did it. The title is informative and relevant, and the references are relevant and recent with the inclusion of appropriate key studies.

Introduction/background

The topic of research is clearly defined, but it is not justified by what is already known on the subject.

In particular, the first sentence of the introduction is important to define the theoretical network into which you want to fit, so I suggest you restructure it and mention the source from which you start.

When you talk about "The reporting system of Islamic banks is different from that of conventional banks because of the differences in the objectives and the users' needs as users desire high-quality information to ensure that the banks are following Sharia rules and guidelines" it is important to cite the source from which the information was taken. This condition is often repeated; therefore, I recommend a revision of the introduction to include all sources from which these developments originated. The importance of caring for this aspect is linked to the paragraph where you say "This study is different in that..." which may not be justified to the reader by studies.

Please conclude the introduction with a small summary of the structure that will follow in the following chapters of the document.

The theoretical section takes up the most important themes for structuring the paper's argument but has two important limitations.

The first is that I would also dedicate a chapter to Islamic Banking. The second is related to the centrality of the topic in literature review. In literature review you must put more references in the performance chapter and in the title, you talk about accounting performance while in literature review you talk about performance with reporting. It is advisable to reorganize the chapter

Methods

In this section the subject selection process is clear, and the variables are well defined and measured appropriately. I have nothing to suggest for this chapter

Results

For this section the data are presented in an appropriate form and the tables and figures are relevant and clearly presented. Units of measurement, rounding and number of decimals and headings, columns and rows shall also be correctly and clearly labelled. In addition, the text of the results is well linked to the data by clarifying what is a statistically significant result.

Discussion and conclusions

In the section Discussion and Conclusions, the results should be discussed from different points of view and contextualized without being overly interpreted. In addition, responses to the objectives of the study in relation to the results should clearly emerge from the conclusions.

In addition, there must be a specific development to the response to the research question.

It is necessary to include the discussion chapter and expand the chapter of conclusions as it is not exhaustive. Consider whether to make a single chapter of discussion and conclusions or separate ones, but there must be.

Overall

The design of the study was appropriate to achieve the objective and the data are significant. It was not well explained what this study added to what was already known about this topic and what were the main weaknesses of this article.  Besides the review of the first chapters the problem is the lack of the last ones, so the paper needs to be rewritten. For these reasons, major revisions are needed to make this article publishable.

Author Response

 Response to Reviewer 1 Comments

The researcher would like to thank the reviewer for the valuable comments. This is to confirm that all the comments and concerns of the reviewer have been fixed in the revised copy of the research paper. It gives great pleasure to inform you that, I really gained great benefits from the reviewer’s comments.  Actually, major changes have been done and I preferred to include the changes on the manuscript. The changes are highlighted in a brown or light brown colour on the original document.

Point 1: The topic of research is clearly defined, but it is not justified by what is already known on the subject. In particular, the first sentence of the introduction is important to define the theoretical network into which you want to fit, so I suggest you restructure it and mention the source from which you start. It is important to cite the source from which the information was taken. Therefore, I recommend a revision of the introduction to include all sources from which these developments originated.

Response 1: To justify the selection of the topic and its significance, the author selects the most relevant work done so far in the literature, the context of Islamic banks, and the research topic. The introduction had been restructured to reflect what is done, where is the gap, and what is the contribution of the study in bridging the gap. Please refer to page 1 (line 1 to line 16) and page 2 (Line 8 to line 14). The author has revised the introduction section and included all the sources and citations. The changes are highlighted in brown colour.

Point 2: Please conclude the introduction with a small summary of the structure that will follow in the following chapters of the document.

Response 2: Yes, this point was not included. The revised version includes a 4-line paragraph about the structure of the manuscript on page 2 just before objectives of the study. It is highlighted in brown colour.

Point 3: The theoretical section takes up the most important themes for structuring the paper's argument but has two important limitations. The first is that I would also dedicate a chapter to Islamic Banking. The second is related to the centrality of the topic in literature review. In literature review you must put more references in the performance chapter and in the title, you talk about accounting performance while in literature review you talk about performance with reporting. It is advisable to reorganize the chapter.

Response 3: The author recognizes this comment and revises the section taking major actions on the document.

The first limitation about literature on Islamic Banking: The author would like to be very close to the research objectives and added the most relevant literature about Islamic Banking. In this part, the author added around 24 lines to the literature section (highlighted in brown colour). The added literature is briefly about the nature, growth, contribution of the GCC Islamic banks, reporting system, and reporting quality.

The second limitation: Centrality of the topic in the literature: The author considers this point and relates every paragraph or concept to the quality of disclosure, profitability, and earnings quality.

The performance section has been re-written and includes more literature on Islamic Banking performance. For details, please refer to section 2.2 on page 4 and page 5. The changes are highlighted in brown colour.

Point 4: In the section Discussion and Conclusions, the results should be discussed from different points of view and contextualized without being overly interpreted. In addition, responses to the objectives of the study in relation to the results should clearly emerge from the conclusions. In addition, there must be a specific development to the response to the research question.

It is necessary to include the discussion chapter and expand the chapter of conclusions as it is not exhaustive. Consider whether to make a single chapter of discussion and conclusions or separate ones, but there must be.

Response 4: More analysis and discussion are added to the manuscript to make the findings more useful. The author is also concerned with over interpretation. Changes have been conducted on pages 9, 10, and 11.

The author preferred to make a separate section for conclusion and it is expanded to include research objectives and relates them to the findings of the study. Furthermore, adds information about implications and limitations of the study as part of the conclusion, which is completely revised. This change is available on page 12.

Overall, all the points raised by the reviewer have been considered and the paper is almost 80% revised or re-written. All possible improvements have been conducted, English proof reading, and writing skills have been improved.

Reviewer 2 Report

Dear author(s),

The manuscript is a good enough one because it analyses a modern issue, but it is mandatory you make some important improvements regarding the literature review section and the methodology + results sections.

Please take into considerations the following remarks and implement them one by one:

  1. In your paper, there are some texts from another articles:
    1. Page 3: “…as applied by Leuz et al. (2003), measures the variability of earnings by calculating the ratio of the standard deviation of operating earnings to the standard deviation of cash from operations.” This text comes from https://www.emerald.com/insight/content/doi/10.1108/02686900510625334/full/html
    2. Page 3: “…on earnings surprise as reflected in the beginning balance of net operating assets relative to sales.” This text comes from https://www.emerald.com/insight/content/doi/10.1108/02686900510625334/full/html
    3. Page 4: “…The independent variable of the study is the quality score, which is composed of the six elements of the QCs of accounting information. The study applies a manual content analysis to the annual reports to score all items of the QCs. To control for biased interpretations, two research assistants provided individual second assessment scores for the QCs to determine the items’ measurement scores. To guarantee the reliability of the scores, Krippendorff’s alpha coefficient is used.” This text seems to be very similar with the text from the following paper: https://www.ersj.eu/dmdocuments/2017-xx-4-b-53.pdf?cv=1 (Page 715 the second paragraph)
    4. Page 7: “To evaluate the outer model, the reliability of each item is measured by the cross-loadings, and it is found that the values of composite reliability are high on their respective constructs, which means that each factor loading is greater than the cut-off value of 0.70. This also proves that the reliability of each variable is good, providing reinforcement to the allocation for each item on the specified latent construct. The study compares the square root of each AVE, and their values are within 0.56 and 0.65, which indicates very small differences. It is an acceptable validity score in that all the AVE values are above 0.50.” This text is very similar to https://iopscience.iop.org/article/10.1088/1742-6596/890/1/012163/pdf (section 4.1)

So, first of all, you must reformulate this paragraphs.

  1. The quality of the literature review must be improved. So, please include in your literature review section the following good quality articles:
  • https://doi.org/10.3390/sym11121449

  • http://www.ecoforumjournal.ro/index.php/eco/article/view/884

  • http://journals.univ-danubius.ro/index.php/oeconomica/article/view/1423/1319

  • https://www.researchgate.net/publication/228319519_The_Adoption_of_Electronic_Banking_Services_in_Developing_Countries_-_The_Romanian_Case

  1. At page 6 in your paper, paragraph no 4, you tell that R=0.10 is acceptable. Please put a bibliographic reference to argue this value.

  1. In the section 3, please specify the software tool you used (e.g.: Amos, Spss, Excel etc) to compute the calculations.

  1. At page 7 in your paper: “Cronbach’s alpha is the most common measurement for internal consistency and composite reliability. It measures the reliability based on the interrelationship of the observed items’ variables. The internal consistency among the variables is achieved and acceptable once the values of Cronbach’s alpha reach between 0.60 to 0.70, as shown in the first column of Cronbach’s alpha in table 3 above.” But in the table 3 there is only one variable between 0.60-0.70. Please correct.

  1. Page 8, below table 7: “The examination of the individual model paths shows a significant positive relationship between EQ and most of the QCs of financial information.” But in the table, the first variable is not in a positive relation”. Please correct.

  1. Page 8, last paragraph: “Figure 2 shows the model components and their relationships. Out of the six quality variables, only understandability is below 1.96 in supporting EQ, while all the remaining variables are above 1.96”. But in the figure 2 from page 9, Relevancy is also below 1.96. Please correct the sentence.

Author Response

Response to Reviewer 2 Comments

The researcher would like to thank the reviewer for the valuable comments. This is to confirm that all the comments and concerns of the reviewer have been fixed in the revised copy of the research paper. It gives great pleasure to inform you that, I really gained great benefits from the reviewer’s comments.  Actually, major changes have been done and I preferred to include the changes on the manuscript. The changes are highlighted in a brown or light brown colour on the original document.

Point 1: The manuscript is a good enough one because it analyses a modern issue, but it is mandatory you make some important improvements regarding the literature review section and the methodology + results sections.

Response 1: The author believes in continuous improvements and appreciates this recommendation. the author reviews the most relevant work done so far in the literature and on that base, revised the introduction to reflect what is done, where is the gap, and what is the contribution of the study in bridging the gap. The author would like to be very close to the research objectives and added the most relevant literature about Islamic Banking. In this part, the author added around 24 lines to the literature section. The results section has been completely revised and the conclusion is expanded based on the results of the study. The changes are highlighted in brown or orange colour.

Point 2: Please take into considerations the following remarks and implement them one by on.    In your paper, there are some texts from another articles:

Response 2: It is really noticed that some texts are from other articles. Some of them may not be written in a proper citation forms but I would like to assure that there is no intention of plagiarism.

  • Page 3: To avoid any kind of plagiarism this text has been re-phrased and written in proper form as on page 4, its source is documented, and highlighted in orange colour.
  • Page 3: This text has been re-phrased and written in proper form as on page 4 and highlighted in orange colour.
  • Page 4: It is very difficult to avoid some common words. This is to assure you that the text has been revised and written in proper form as on page 6 and highlighted in orange colour.
  • Page 7: This text has been re-phrased and written in proper form as on page 4 and highlighted in orange colour.

All previous comments are well documented and the sources are included in the list of references.

Point 3: The quality of the literature review must be improved. So, please include in your literature review section the following good quality articles:

Response 3: The author revises the literature section and improve it by including all studies relevant to the topic and the context. This change is available in the introduction and literature sections. Thanks to the reviewer for suggesting valuable articles. The author includes the relevant citations from the mentioned articles and cited them in the list of references. They suggested papers and citations are written in orange colour.

The performance section has been re-written and includes more literature on Islamic Banking performance. For details, please refer to section 2.2 on page 4 and page 5. The changes are highlighted in brown colour.

Point 4: 3. At page 6 in your paper, paragraph no 4, you tell that R=0.10 is acceptable. Please put a bibliographic reference to argue this value.

Response 4: The value of R2 is one of the critical issues in statistics. Since R2 value is adopted in various research discipline, there is no standard guideline to determine the level of predictive acceptance. Henseler (2009) proposed a rule of thumb for acceptable R2 with 0.75, 0.50, and 0.25 are described as substantial, moderate and weak respectively. Another opinion accepts R2 at 10% in humanities and social sciences. I followed this opinion and it cited on page 8 of the manuscript.

Point 5: 4. In the section 3, please specify the software tool you used (e.g.: Amos, Spss, Excel etc) to compute the calculations.

Response 5: The study follows the Neural Networks as part of the MATLAB software. It is described in section 3.1, on page 6, the paragraph before the last.

Point 6: At page 7 in your paper: “Cronbach’s alpha is the most common measurement for internal consistency and composite reliability. It measures the reliability based on the interrelationship of the observed items’ variables. The internal consistency among the variables is achieved and acceptable once the values of Cronbach’s alpha reach between 0.60 to 0.70, as shown in the first column of Cronbach’s alpha in table 3 above.” But in the table 3 there is only one variable between 0.60-0.70. Please correct.

Response 6: Yes, the author recognizes this mistake and it is corrected and properly written on page 9. The expression was not properly written, it was meant to accept 0.7 or even between 0.6 and 0.07. Thank you for this notification.

Point 7: Page 8, below table 7: “The examination of the individual model paths shows a significant positive relationship between EQ and most of the QCs of financial information.” But in the table, the first variable is not in a positive relation”. Please correct.

Response 7: Thank you. I appreciate such accurate comments. The word positive is now removed. Noticed and corrected on page 10.

Point 8: Page 8, last paragraph: “Figure 2 shows the model components and their relationships. Out of the six quality variables, only understandability is below 1.96in supporting EQ, while all the remaining variables are above 1.96”. But in the figure 2 from page 9, Relevancy is also below 1.96. Please correct the sentence.

Response 8: The comment is appreciated such. The analysis has been accurately written and the information is revised on page 10 and highlighted in orange colour.

Overall, all the points raised by the reviewer have been considered and the paper is fully revised to include all possible improvements, English proof reading, and writing skills have been improved as well.

Round 2

Reviewer 1 Report

Abstract, title and references

The purpose is clear. It's clear what the study discovered and how they did it. The title is informative and relevant and the references are relevant and recent with the inclusion of appropriate key studies.

Introduction/background

It is clear what is already known on this subject. The research question is clearly outlined and is justified by what is already known on the subject.

Methods

The subject selection process is clear. Variables are defined and measured appropriately and the study method is valid and reliable.

Results

The data shall be presented in an appropriate manner. The tables and figures shall be relevant and clearly presented. The units of measurement, rounding and number of decimals shall be appropriate. Headings, columns and rows shall be correctly and clearly labelled. Categories are grouped appropriately and the text of the results is added to the data and is not repetitive. it is clear what is a statistically significant result.

Discussion and conclusions

The results are discussed from several points of view and contextualized without being overly interpreted. The conclusions meet the objectives of the study and are supported by references and results.

Overall

The design of the study was adequate to meet the objective and is consistent with itself.

Reviewer 2 Report

Dear Author(s),

The revised version of the manuscript is ok now.

Congratulations for your work!

Good luck!